# Rapid electrochemical detection of coronavirus SARS-CoV-2

Thanyarat Chaibun[1], Jiratchaya Puenpa[2], Tatchanun Ngamdee[3], Nimaradee Boonapatcharoen[4], Pornpat Athamanolap[1], Anthony Peter O'Mullane [5], Sompong Vongpunsawad[2], Yong Poovorawan[2], Su Yin Lee[6,7] & Benchaporn Lertanantawong [1✉]

Coronavirus disease 2019 (COVID-19) is a highly contagious disease caused by severe acute respiratory syndrome coronavirus 2 (SARS-CoV-2). Diagnosis of COVID-19 depends on quantitative reverse transcription PCR (qRT-PCR), which is time-consuming and requires expensive instrumentation. Here, we report an ultrasensitive electrochemical biosensor based on isothermal rolling circle amplification (RCA) for rapid detection of SARS-CoV-2. The assay involves the hybridization of the RCA amplicons with probes that were functionalized with redox active labels that are detectable by an electrochemical biosensor. The one-step sandwich hybridization assay could detect as low as 1 copy/µL of $N$ and $S$ genes, in less than 2 h. Sensor evaluation with 106 clinical samples, including 41 SARS-CoV-2 positive and 9 samples positive for other respiratory viruses, gave a 100% concordance result with qRT-PCR, with complete correlation between the biosensor current signals and quantitation cycle (Cq) values. In summary, this biosensor could be used as an on-site, real-time diagnostic test for COVID-19.

[1] Biosensors Laboratory, Department of Biomedical Engineering, Faculty of Engineering, Mahidol University, Nakhon Pathom, Thailand. [2] Center of Excellence in Clinical Virology, Faculty of Medicine, Chulalongkorn University, Bangkok, Thailand. [3] Department of Biotechnology, School of Bioresources and Technology, King Mongkut's University of Technology Thonburi, Bangkok, Thailand. [4] Pilot Plant Development and Training Institute (PDTI), King Mongkut's University of Technology Thonburi, Bangkok, Thailand. [5] School of Chemistry and Physics, Queensland University of Technology (QUT), Brisbane, QLD, Australia. [6] Faculty of Applied Sciences, AIMST University, Bedong, Kedah, Malaysia. [7] Centre of Excellence for Omics-Driven Computational Biodiscovery (COMBio), AIMST University, Bedong, Kedah, Malaysia. ✉email: benchaporn.ler@mahidol.ac.th

The global outbreak of coronavirus disease 2019 (COVID-19) is caused by the rapid spread of severe acute respiratory syndrome coronavirus 2 (SARS-CoV-2)[1]. SARS-CoV-2 has been classified as a beta coronavirus with high nucleotide sequence homology to two severe acute respiratory syndrome (SARS)-like bat coronaviruses and moderate homology with the Middle East respiratory syndrome coronavirus CoV[2,3]. The virus genome is a single positive-stranded RNA of ~30,000 bases in length, which encodes for 10 genes, including the 5′ untranslated region, replicase complex (*orf1ab*), spike (*S*), envelope (*E*), membrane (*M*), and nucleocapsid (*N*) structural proteins, 3′-untranslated region, and several unidentified non-structural open reading frames[4].

The mode of virus transmission includes droplet, contact, airborne, fomite, fecal-oral, and bloodborne transmissions, which exacerbates the rapid spread of the virus[5]. A person infected with SARS-CoV-2 can either remain asymptomatic or show non-specific clinical symptoms such as fever, cough, or shortness of breath. Even during the incubation period, an infected person is highly contagious and can spread the virus to a non-infected person[6]. Thus, rapid diagnostic testing for SARS-CoV-2 at a large scale is crucial for virus detection, surveillance, and swift management of outbreaks[7].

There are two types of diagnostic tests for COVID-19, serological and viral nucleic acid tests. Serological testing detects the presence of antibodies produced by an individual due to exposure to the virus or detection of antigenic viral proteins in the infected individuals. An antibody test should not be used to diagnose someone with an active infection and may give false-negative results. Rapid antigen tests for SARS-CoV-2 are relatively inexpensive and give immediate results. However, these tests are generally less sensitive than nucleic acid-based tests. At the early stage of infection, the human immune system may not be active and thus cause false-negative diagnosis[8].

Therefore, for the accurate diagnosis of an active COVID-19 infection, viral nucleic acid testing should be used. At present, quantitative reverse transcription polymerase chain reaction (qRT-PCR) is the gold standard for the diagnosis and confirmation of SARS-CoV-2 infection. Examples of commercially available qRT-PCR test kits are the Xpert Xpress SARS-CoV-2 test[9], CDC 2019- novel coronavirus Real-Time RT-PCR Diagnostic Panel[10], ExProbeTM SARS-CoV-2 Testing Kit[11], Abbott RealTime SARS-CoV-2 RT-PCR Kit[12], PerkinElmer® New Coronavirus Nucleic Acid Detection Kit[13], and TaqPath COVID-19 Combo Kit[14]. However, there are disadvantages with these methods as they are time-consuming, requiring a specialized laboratory setting with expensive instruments and trained personnel[15].To overcome these drawbacks, various isothermal nucleic acid amplification assays such as recombinase polymerase amplification (RPA)[16] and loop-mediated isothermal amplification (LAMP)[17–20], deoxyribonucleic acid (DNA) nanoscaffold-based hybrid chain reaction[21], and nucleic acid sequence-based amplification[22], have been developed to overcome the need for sophisticated thermal cycling equipment associated with qRT-PCR.

An isothermal amplification method known as rolling circle amplification (RCA) has also been widely used for nucleic acid testing[23]. The RCA assay involves the amplification of DNA or ribonucleic acid (RNA) primers that are annealed to a circular DNA template using DNA or RNA polymerases[24]. The RCA amplicon is a concatemer containing multiple repeats of sequences that are complementary to the circular template. RCA is able to produce amplicons ~$10^9$-fold within 90 min with minimal reagents[25]. Owing to its isothermal nature, RCA can be performed using a simple water bath or heating block. A significant advantage of RCA is that the detection of the amplicons

can be undertaken using an electrochemical biosensor, which enables rapid, quantitative results to be obtained either in the laboratory or more importantly, in a field setting. This allows for rapid, widespread deployment of testing kits to outbreak areas or remote regions where laboratory facilities do not exist or are difficult to access. This approach will be of particular benefit to developing countries. An RCA assay for the 2003 SARS-CoV has been reported, however, detection of the RCA amplicons relied on cumbersome and time-consuming gel electrophoresis[26]. Recently, an RCA-based real-time optomagnetic detection of synthetic complementary DNA of SARS-CoV-2 was reported[27].

In this study, we describe an electrochemical biosensor based on multiplex RCA for the rapid detection of the *N* and *S* genes of SARS-CoV-2 from clinical samples (Fig. 1). The assay involves sandwich hybridization of RCA amplicons with probes that are functionalized with redox-active labels, which are subsequently detected by differential pulse voltammetry (DPV). The assay could detect as low as 1 copy/μL of viral *N* or *S* genes in <2 h. Clinical samples were also used to evaluate the performance of the assay, which was found to be in agreement with qRT-PCR results.

## Results

### Characterization of the silica core and redox dye-incorporated silica nanoparticles.
Two redox dyes, methylene blue (MB) and acridine orange (AO) were coated onto silica nanoparticles (SiNPs) through surface-reactive functional groups. Figure 2a–c displays the scanning electron micrographs (SEM) of silica core particles, silica with a redox-dye layer, denoted as Silica-methylene blue (SiMB) and Silica-acridine orange (SiAO). The size of the SiNPs after coating with the redox-dye increased compared with the silica core, which is similar to previous report by Cheeveewattanagul[28]. In addition, we performed layer-by-layer modification with two polyelectrolytes, which are positively charged poly(allylamine)hydrochloride (PAA) and negatively charged poly(sodium 4-styrene) sulfonate (PSS), onto SiMB and SiAO particles. The absorbed PSS left a net negative charge on the surface of the SiNPs, which facilitated the binding with the avidin linker.

### Detection of RCA amplicons by agarose gel electrophoresis.
In the presence of the target gene, the circular DNA template hybridizes to the target gene, causing the 5′ end of the circular DNA template to be juxtaposed to its 3′ end (Fig. 2d). Following ligation, the Padlock DNA serves as a template for amplification by phi29 DNA polymerase to produce RCA amplicons, which are long DNA amplicons containing hundreds of tandem repeats of the Padlock DNA complementary sequence. This sequence lies in the universal capture probe and gene-specific silica-tagged reporter probe (Si-RP)-binding regions. Figure 2e shows the successful amplification of the N and S genes visualized using agarose gel electrophoresis. Within 30 min of amplification, high molecular weight RCA amplicons were formed, which did not migrate out of the wells because of their large size. The electrophoresis result of circular DNA templates in the absence of the target gene and without amplification is shown in Supplementary Fig. 1.

### Performance of the sandwich and one-step hybridization.
The performance of the step-wise sandwich and one-step hybridization strategies were compared using 1 pM of linear target for the *N* gene. Step-wise sandwich hybridization involved the sequential hybridization of the capture probe-conjugated magnetic bead particle (CP-MNB) to the target and the CP-MNB-target to the silica-reporter probe (Si-RP), with washing steps between each

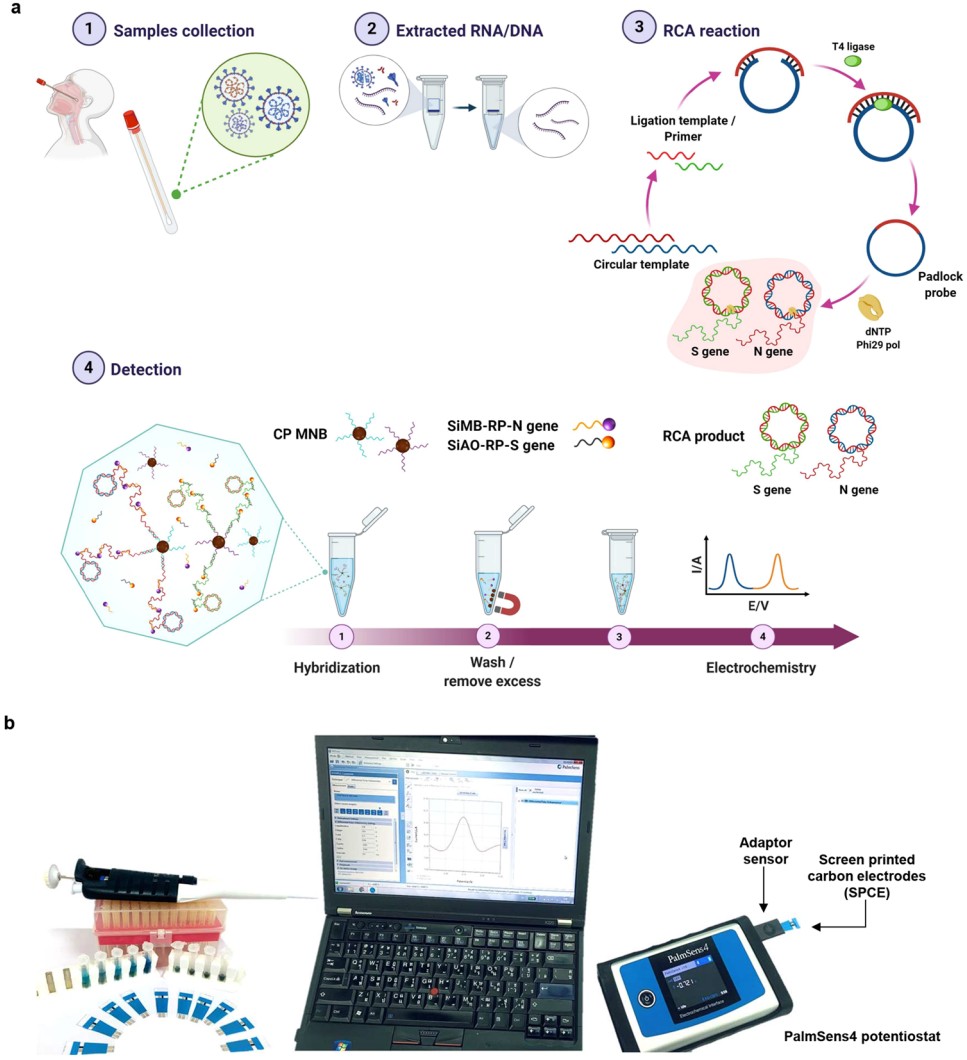

**Fig. 1 Overview of the detection platform. a** Detection workflow of SARS-CoV-2 from clinical samples using the electrochemical biosensor with RCA of the *N* and *S* genes. **b** The detection setup for electrochemical analysis by using a portable potentiostat device connected to a laptop.

hybridization. In the one-step strategy, CP-MNB, Si-RP, and the target were mixed in a single hybridization step, followed by a single washing step (Fig. 3a). We found no significant difference ($p > 0.05$; two-tailed Student's *t* test) in the current signal between the step-wise sandwich and the one-step hybridization strategies (Fig. 3b), Therefore, the one-step hybridization strategy was used for the optimized assay as it was easier, faster, and reduced pipetting steps. This also minimizes contamination, which is a key concern for field-based testing. The use of magnetic beads coated with streptavidin enables easy manipulation of the target nucleic acid during isolation and purification using a magnetic field[29,30]. The complementary DNA (cDNA) or RNA target will be specifically hybridized to the biotinylated probe on the magnetic beads via hydrogen bonding, permitting their precise targeting. Moreover, this approach has become a valuable method because of their low cost, compatibility with liquid sample, and high surface area for hybridization[31].

**Assay sensitivity**. For both the *S* and *N* genes, the increase in the electrochemical signals were positively correlated to the increase in the gene copy number (Fig. 3 c, d, e). Surprisingly, even though the two genes were detected with different redox dyes (MB for *N* gene and AO for *S* gene), the strength of the current signals obtained from both genes were very similar. The similarity of the signals was

attributed to the same mechanism of the redox reaction (two-electron oxidation) for both dyes used in this study (Supplementary Fig. 2)[28,32]. The detection limit of both the *N* and *S* genes was 1 copy/μL, with a linear range of 1 to $1 \times 10^9$ copies/μL. The correlation of current signal for *N* and *S* genes was 99%. The plasmid concentrations in molarity are shown in Supplementary Table 1.

**Assay specificity**. COVID-19 and influenza and are both contagious respiratory illnesses. Although they are caused by different viruses, the symptoms of COVID-19 and influenza are very similar and may be hard to distinguish based on clinical symptoms alone. Therefore, a diagnostic test that can distinguish the influenza virus from COVID-19 is necessary to avoid misdiagnosis. In the specificity testing, 1 pM of target sequences of Influenza A (IAV) and Influenza B (IVB) viruses were included as non-complementary targets, where the sequence alignment is shown in Fig. 3f. In addition, 1 pM of linear targets with two bases mismatch were also included. A current signal that was equal or $\geq +3$ standard deviations (3 SD) above the mean of the blank (background) signal was considered a positive result. All complementary targets for both the *N* and *S* genes yielded positive results, while the non-complementary targets (IAV, IBV, IAV + IBV, mismatch) were all negative (Fig. 3g). Moreover, the

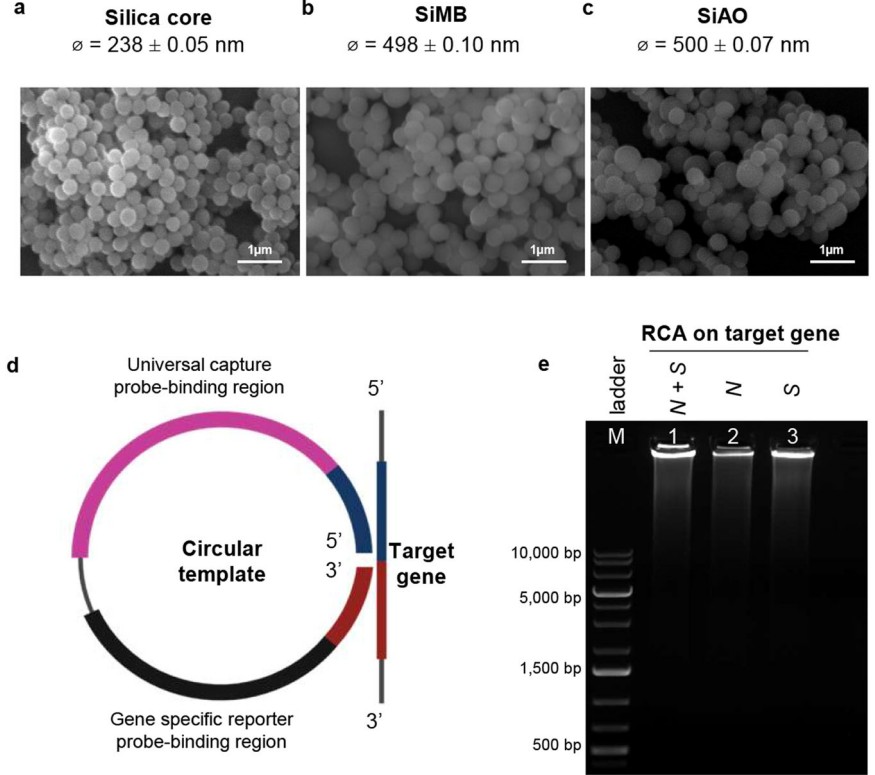

**Fig. 2 Visualization of the silica-redox dye and RCA amplicons. a** SEM image with 1 μm scale bar of silica core, **b** silica-methylene blue (SiMB), and **c** silica-acridine orange (SiAO) composite particles with diameter sizes ($\phi$) which are presented as mean values ± standard deviation (SD), $n = 10$. **d** Target gene, universal capture probe, and gene-specific reporter probe binding regions on the circular DNA template. **e** Gel representation of RCA amplicons visualized on 0.8% agarose gel electrophoresis ($n = 3$). Lane M indicates the 1 kb DNA ladder, while lanes 1, 2, and 3 indicate the RCA amplicons of both *N* and *S* genes, *N* gene only, and *S* gene only, respectively. Source data are provided as a Source Data file.

discrimination between perfectly complementary targets and two bases mismatch targets demonstrates that the assay is highly specific. Therefore, this assay could be useful for detecting SARS-CoV-2, even in the presence of co-infection with other viruses that manifest similar respiratory symptoms.

**Performance of the assay for detection of SARS-CoV-2 from clinical samples**. RNA and cDNA samples prepared from clinical samples were used as the template in RCA, using the one-step hybridization method and electrochemical detection. All 41 samples (11 RNA and 30 cDNA) prepared from SARS-CoV-2-positive clinical samples yielded positive results, while the 65 samples (40 RNA, 25 cDNA) prepared from SARS-CoV-2-negative clinical samples recorded negative results (Fig. 4). This result was concordant with the qRT-PCR results. The differential pulse voltammogram (DPV) data for each of these tests is shown in Supplementary Fig. 3.

**Discussion**
RCA is a robust and technically simple, isothermal technique for in vitro DNA amplification. RCA uses strand displacing phi29 DNA polymerase to continuously amplify the circular nucleic acid template[33]. The unique concatenated DNA strand that is produced as the amplification product has multiple binding sites for the RP to hybridize[34] (~$10^2$–$10^3$ repeats of complementary sequence, and even up to $10^5$ in some cases can be achieved on the RCA amplicons)[23]. In addition, the circular DNA can bind both DNA and RNA targets which advantageously eliminates the need to perform a reverse transcription step, unlike other isothermal methods such as RPA and LAMP. The use of T4 DNA

ligase enables the joining of adjacent blunt end termini of DNA/RNA hybrids[35].

In comparison with PCR-based assays, RCA can be performed under isothermal conditions with minimal reagents and avoids the generation of false-positive results. RCA assay is also less complicated compared with other isothermal amplification methods, such as a transcription-based system, strand displacement approach, invasive-cleavage reaction, or loop-mediated technology. RCA can be performed with a minimal pre-optimization step, hence it can be readily employed by non-skilled users[23].

Target amplification by RCA followed by electrochemical biosensor detection requires three steps of target recognition and hybridization to its complementary sequence. First, the gene target must hybridize with the complementary sequence in the circular DNA, followed by ligation of the circular DNA to produce a Padlock DNA. After amplification of the Padlock DNA by phi29 DNA polymerase, RCA amplicons are produced, which are long repeats of the complementary sequence of the Padlock DNA. Next, the RCA amplicons are bound by the reporter and capture probes, followed by electrochemical detection of the redox-active dye. This strategy ensures high specificity is achieved with the assay. Moreover, utilization of magnetic capture and separation of targets from non-targets reduces the chances of carry-over contamination and pipetting error.

The World Health Organization (WHO) recommends that for laboratory confirmation of cases by nucleic acid testing in areas with no known COVID-19 virus circulation, the test must be positive for at least two different targets on the COVID-19 virus genome. In areas where COVID-19 virus circulation has been established, nucleic acid testing for a single discriminatory target

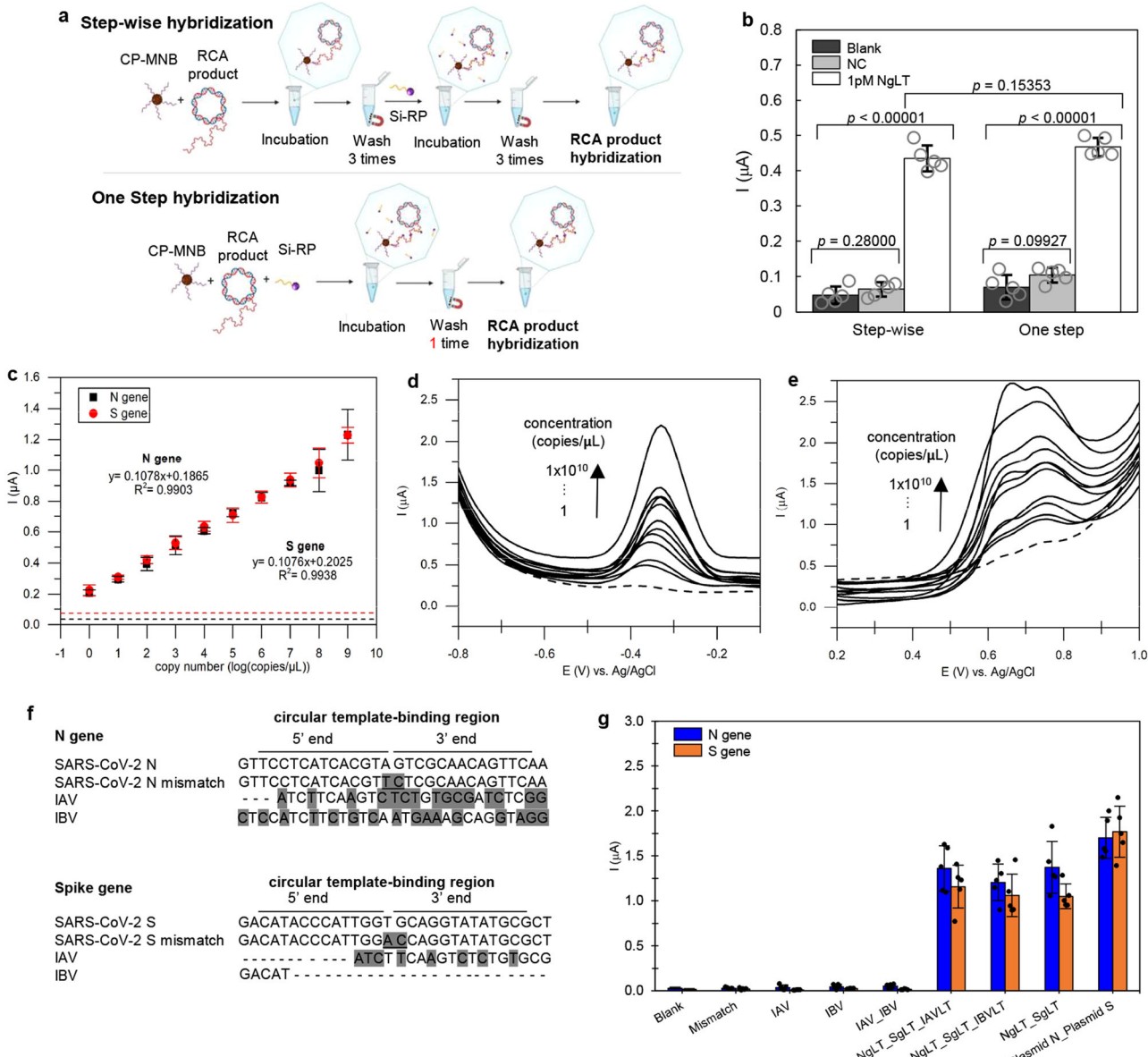

**Fig. 3 Assay performance. a** Comparison of the step-wise (step-by-step) and one-step sandwich hybridization procedures using the *N* gene as the target. **b** Step-wise and one-step hybridization strategies produce current signals that are not significantly different ($p > 0.05$). The two-tailed Student's *t* test was used to compare differences between two groups. All current responses are presented as mean values±standard deviation (SD), $n = 5$. The corresponding data points (as dot plots) are overlaid. **c** Sensitivity assay for *N* and *S* genes shows a positive correlation in current response to the copy number for both genes. All current responses are presented as mean values ±SD, $n = 5$. **d** Differential pulse voltammograms showing the increase in the current signal as the concentrations of the *N* gene increased. Dashed lines show the blank signal. **e** Differential pulse voltammograms of the *S* gene showing the increase in the current signal as the concentrations of the *S* gene increased. Dashed line shows the blank signal. **f** Multiple sequence alignment of the *N* and *S* genes target sequences with mismatch and non-complementary target sequences. Dark shaded areas are the non-complementary sequences to target gene, while the underlined bases are the mismatch bases. **g** The specificity of the assay with *N* (blue bar) and *S* (orange bar) genes, tested with perfect complementary targets (*Ng*, *Sg*, Plasmid N, Plasmid S), mismatch DNA target (Mismatch), and non-complementary targets (IAV, IBV) and the corresponding data points (as dot plots) are overlaid. Source data are provided as a Source Data file.

is considered sufficient[36]. Therefore, in this study, we performed multiplex RCA for the simultaneous amplification of two genes, which are the *N* and *S* genes encoding the nucleocapsid and spike proteins, respectively. We designed circular DNA templates with the same capture probe binding sequence for both *N* and *S* genes so that the CP-MNB could bind to the RCA amplicons of both genes. The RCA amplicons containing both *N* and *S* genes were electrochemically detected using the respective redox-labeled RP.

Other assays for SARS-CoV-2 detection also targeted the *N* and *S* genes using qRT-PCR and conventional RT-PCR[37–39]. In

addition, some diagnostic tests have been designed to target the *E* gene to broadly detect coronavirus infections (bat SARS-like coronavirus and SARS-CoV)[20,40,41]. Besides these genes, other commonly used gene targets for detection of SARS-CoV-2 are the *ORF1ab*[17] and *RdRp*[42] genes.

RNA viruses such as SARS-CoV-2 have a high mutation rate, which contributes to its adaptation[43]. SARS-CoV-2 genomic variations have been associated with differences in the mortality rate of COVID-19. Mutations in the *RdRP*[43], *ORF1ab*[44], *N*[45], and *S*[46] genes have been identified. Therefore, it is recommended that

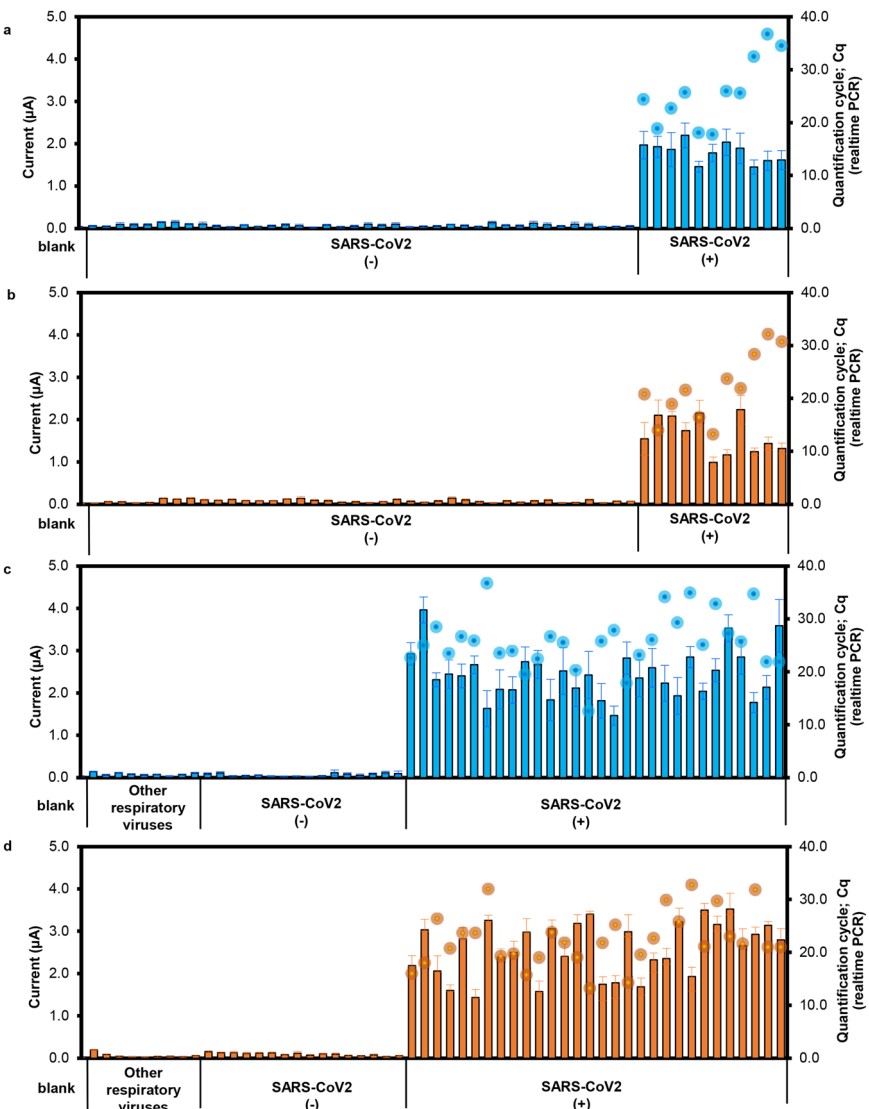

**Fig. 4 RCA with rapid electrochemical detection of N and S genes of SARS-CoV-2 in clinical samples. a** The detection of *N* gene in 51 RNA samples. **b** The detection of *S* gene in 51 RNA samples. **c** The detection of *N* gene in 55 cDNA samples. **d** The detection of *S* gene in 55 cDNA samples. All samples used for the evaluation were prepared from nasopharyngeal swabs. The current signals from electrochemical measurements of the *N* gene (blue bar graph) and *S* gene (orange bar graph) are shown compared with the Cq result from qRT-PCR (blue dots for *N* gene and orange dots for *S* gene. All current responses are presented as mean values±standard deviation (SD), $n = 5$, whereas Cq values are from single measurements).

at least two targets are included in a SARS-CoV-2 diagnostic test to reduce the possibility of false-negatives due to mutations in the target genes. On the other hand, the ability of the assay to discriminate two bases mismatch could be useful for viral mutation studies. Therefore, wild-type and mutant strains could in principle be distinguished using probes tagged with different redox dyes as demonstrated in this study. Another advantage of our assay is the limit of detection (LoD), which showed that it is highly sensitive and comparable to commercially available test kits and existing biosensors (Table 1). Our assay's LoD of 1 copy/ μL for the *N* gene is comparable, if not slightly better, than CDC's RT-PCR assay of 3.2 copies/μL[10,15].

The application of the optimized assay for the diagnosis of COVID-19 was evaluated with 106 clinical samples. This included samples that were SARS-CoV-2 negative but positive for other respiratory viruses. The ability of a test to distinguish between SARS-CoV-2 and other respiratory viruses is important because the clinical symptoms of COVID-19 and influenza are very similar, however, the approach in management and control of the

disease is different. Clinical diagnosis is further complicated in cases of co-infection with SARS-CoV-2 and influenza[47]. A multiplex test that can detect SARS-CoV-2, influenza, and other respiratory viruses simultaneously would save time and provide more accurate information to clinicians and public health officials[48]. At present, we have only tested multiplex RCA with two genes, however, it is possible to include additional gene targets for influenza and other respiratory viruses with additional optimization.

A recent study reported that the viral loads of SARS-CoV-2 in clinical samples by RT-PCR ranged from 641 copies/mL to $1.34 \times 10^{11}$ copies/mL, with a median of $7.99 \times 10^4$ in throat samples and $7.52 \times 10^5$ in sputum samples, and $1.69 \times 10^5$ copies/mL for nasal swab samples taken on day 3 of post-onset of symptoms. The average viral load after early onset was $>1 \times 10^6$ copies/mL. In death cases, the viral load was $1.34 \times 10^{11}$ copies/mL detected from sputum samples collected on day 8 of post-onset of symptoms[49]. Therefore, our assay's LoD of 1 copy/μL is lower than the viral load in clinical samples. This shows that our assay fulfills the

**Table 1 Sensitivity of commercially available test kits and existing biosensors for COVID-19 testing.**

| Detection method | Sample | Targets | LOD | Ref. |
|---|---|---|---|---|
| CRISPR technology | Plasma, serum, and throat and nasal swab | - | - | 56 |
| | Nasal swab | *ORF1ab* and *N* gene | 2 copies/sample | 57 |
| | Nasopharyngeal swabs | *E* gene and *N* gene | 10 copies/μL | 20 |
| | - | *S, N* and *Orf1ab* genes | 42 copies/reaction | 58 |
| Colorimetric assay | Oropharyngeal swab | *N* gene | 0.18 ng/μL | 59 |
| Localized surface plasmon resonance | Synthetic cDNA | *RdRp* | $2.26 \times 10^4$ copies | 42 |
| RT-LAMP | Swabs and bronchoalveolar lavage fluid | *orf1ab* | 20 copies/reaction | 18 |
| | | *S* genes | 200 copies/reaction | |
| | Throat swabs | *orf1ab* | 20 copies /25μL | 17 |
| | | *S* gene | 2 copies/25μL | |
| | | *N* gene | 2 copies/25μL | |
| | Nasal swabs | *N* gene | 100 copies | 19 |
| DNA nanoscaffold-based hybrid chain reaction | Synthetic RNA | conserved region | 0.96 pM | 21 |
| RCA | Synthetic cDNA | *RdRp* | 0.4 fM | 27 |
| | Nasopharyngeal swab | *N* gene and *S* gene | 1 copy/μL | This work |

requirement for sensitivity and could potentially be used to diagnose COVID-19 in the early stages of the illness when the viral load is still low. It is known that many factors can affect test performance and cause false-negative results. Several studies have shown that the viral load varies from specimen types, collection methods, and time of collection[5,50]. Therefore, a robust diagnostic test with high sensitivity could reduce the chances of false negatives caused by low recovery of the virus from real samples.

The detection of RCA amplicons using a USB or battery-powered, portable Palmsens4 potentiostat makes it easy to conduct the test in the field or in a community setting. Indeed, electrochemical based detection using techniques such as DPV, is highly sensitive, quantitative, cost-effective, and compatible with multiplexing. The test can be easily setup in places with limited resources such as in developing countries or in community centers where an outbreak has occurred. We are currently optimizing the RNA extraction process and exploring the use of a smartphone-based biosensor device that will further simplify the test procedure and reduce the sample-to-result turnaround time.

In summary, we have developed an electrochemical biosensor coupled with RCA for the highly sensitive and specific detection of SARS-CoV-2. The high amplification capability of RCA and sensitivity of the electrochemical detection method enabled the detection of the viral *N* and *S* genes in synthetic linear targets as well as clinical samples. The whole assay took under 2 h to complete, from RNA extraction to the detection step, and does not require the use of a thermal cycler. The performance of the assay with clinical samples was comparable to that of RT-qPCR, which is currently the standard for SARS-CoV-2 detection, whereas also demonstrating zero false-positive results. This approach may have a significant impact in places where rapid detection is required to minimize emerging SARS-CoV-2 outbreaks.

## Methods
**Chemicals, reagents, and instrumentation**. Tetraethyl orthosilicate (TEOS), AO, PAA (MW ~56,000), PSS (MW ~70,000), avidin, were purchased from Sigma-Aldrich, USA. MB, 25% ammonia solution, 2-propanal were purchased from Merck, Germany. Dynabeads[TM] MyOne[TM] Streptavidin T1 was purchased from Thermo Fisher Scientific, USA. RCA reagents (deoxyribonucleotide triphosphate (dNTPs), phi29 DNA polymerase were purchased from Integrated DNA technologies, USA. MagLEAD Consumable Kit and magLEAD 12gC instrument were purchased from Precision System Science, Chiba, Japan. For electrochemical measurements, two-electrode screen-printed carbon electrodes (SPCE) were obtained from Quasense, Thailand and the PalmSens4 potentiostat with PSTrace software was obtained from Palmsens, The Netherlands. All the reagent and buffer solutions were prepared with deionized (DI) water (18.2 MΩ).

**Design of circular DNA, primers, and probes**. Circular DNA, primers, and probe sequences were designed to target the *N* and *S* genes of SARS-CoV-2 (GenBank accession number MN908947.3). The oligonucleotide sequences are listed in Table 2. The universal capture probe (RCA-CP) is able to bind the RCA amplicons of both the *N* and *S* genes. The capture, reporter, and blocking probes were tagged with a biotin moiety at either the 3′ or the 5′ end. The mismatched targets for *N* and *S* genes contain two mismatched bases (shown underlined in Table 2). Oligonucleotides were synthesized by Integrated DNA Technologies Pte. Ltd., Singapore.

**Synthesis of monodisperse silica microspheres**. Monodisperse silica microspheres were prepared by the modified Stöber process[51]. First, 1.375 mL of TEOS was added to 9.6 mL of 2-propanol under slow stirring. The solution was heated to 50 °C and a mixture of 0.5 mL of 25% (v/v) ammonia and 1.025 mL of DI water were added with constant stirring for 1 h to form the silica seed. After that, 5 mL TEOS, 227.5 mL of 2-propanol, and 44.5 mL of 8.29% (v/v) ammonia were added into the silica seed solution. Then, 45 mL of TEOS was added to the mixture at a rate of 0.5 mL/min, and the reaction was allowed to continue for an additional 30 min with fast stirring. The total volume and total reaction time were 335 mL and 1 h, respectively. The silica microsphere particles were isolated by centrifugation at $8000 \times g$ for 10 min. The pellet was washed with DI water and centrifuged at $6000 \times g$ for 5 min. The washing step was repeated 4 times. The silica pellet was dried at 105 °C in an oven.

**Incorporation of redox dye onto the silica microspheres**. The redox-active dye was incorporated into the SiNPs using a modified method by Cheeveewattanagul et al.[28]. First, 0.3 g of SiNPs was mixed with 10.9 mL of 2-propanol containing $1.5 \times 10^{-5}$ mol of the redox dye. MB dye was used for *N* gene detection, whereas AO was used for *S* gene detection. The mixture was sonicated at 50 °C for 60 min to ensure good dispersion. Next, under stirring conditions at 40 °C, 0.55 mL of TEOS, 1.5 mL of 25% (v/v) ammonia and 12 mL of DI water were added separately. The reaction was allowed to proceed for another 2 h. The unbound dye was separated by discarding the supernatant after centrifugation at $6000 \times g$ for 5 min. The pellet which was washed with DI water and centrifuged at $6000 \times g$ for 5 min. The washing step was repeated four times. The dye-incorporated silica was dried at 105 °C in an oven. The silica core and dye-incorporated silica were characterized by SEM (JSM-6610LV, JEOL Ltd., Tokyo, Japan).

**Preparation of avidin-coated Si-PAA-PSS particles**. SiMB and DNA conjugation was performed according to Cheeveewattanagul et al.[28]. First, 0.2 mL of PAA (10 mg/mL) was added into a solution containing Si (10 mg/mL) and mixed well via ultrasonication. The mixture was incubated at room temperature for 30 min. Then, the excess polyelectrolyte was removed by centrifugation at $6000 \times g$ for 5 min and washed three times with 0.1 M phosphate-buffered saline (PBS) (pH 7.0). The pellet containing Si-PAA was redispersed in 1 mL DI water. Subsequently, 0.2 mL of PSS (10 mg/mL) was added to the Si-PAA solution, mixed well, and incubated at room temperature for 30 min. After the centrifugation and washing step, the pellet containing Si-PAA-PSS was reconstituted with 1 mL of 10 mM phosphate buffer (PB). Next, 10 μL of avidin (21.14 mg/mL) was added and incubated at 37 °C for 90 min. The avidin-coated Si-PAA-PSS particles were isolated by centrifugation at $6000 \times g$ for 5 min. The pellet was washed with 0.1 M PBS (pH 7.0) and centrifuged at $6000 \times g$ for 5 min. The washing step was repeated three times. The pellet containing avidin-coated Si-PAA-PSS particles (referred to as Si-Avidin) was resuspended in 2 mL of 0.1 M PBS.

**Table 2 Sequences of circular DNA, primers, and probes used in this study.**

| Type | Name | Sequence (5′ → 3′) | Length (bases) |
|---|---|---|---|
| Universal capture probe | CP | CGCAACTGAACTACTTGTCG–biotin | 20 |
| Blocking probe | BP | Biotin-TTTTTTTTTT | 10 |
| *For N gene* | | | |
| Forward primer | Ng_SARS2_F | TCATCACGTAGTCGCAACAG | 20 |
| Reverse primer | Ng_SARS2_R | CAAAGCAAGAGCAGCATCAC | 20 |
| Circular DNA | RCA-Ng | TACGTGATGACGCAACTGAACTACTTGTCGCTGTAGTTCAAGATATCGCGTCCTACCTGTTGCGAC | 66 |
| Reporter probe | Ng-RP | CAAGATATCGCGTCCTAC–biotin | 18 |
| Linear target | Ng-LT | GTTCCTCATCACGTAGTCGCAACA GTTCAA | 30 |
| Mismatched target | Ng-MM | GTTCCTCATCACGTTCTCGCAACA GTTCAA | 30 |
| *For S gene* | | | |
| Forward primer | Sg_SARS2_F | TACCCATTGGTGCAGGTATATG | 22 |
| Reverse primer | Sg_SARS2_R | AGTGTAGGCAATGATGGATTGA | 22 |
| Circular DNA | RCA-Sg | ACCAATGGGTCGCAACTGAACTACTTGTCGCTGTAGTTATTCTGTCATGCGCTCACATATACCTGC | 66 |
| Reporter probe | Sg-RP | ATTCTGTCATGCGCTCAC–biotin | 18 |
| Linear target | Sg-LT | GACATACCCATTGGTGCAGGTATATGCGCT | 30 |
| Mismatched target | Sg-MM | GACATACCCATTGGACCAGGTATATGCGCT | 30 |

**Functionalization of Si-Avidin with the RP**. The conjugation of Si-Avidin with the RP was achieved via the avidin/biotin interaction. The solution containing 0.1 mL of Si-Avidin and 0.3 mL of 0.1 M PBS (pH 7.0) was mixed with 1 µL of 10 µM RP and 99 µL of 10 µM blocking probe (BP). Both RP and BP contain a biotin moiety at the 3′ end, which can bind to avidin. The mixture was incubated at room temperature for 30 min. The DNA-conjugated Si-Avidin was recovered by centrifugation at $6000 \times g$ for 5 min. The pellet was washed with 1 M PBS (pH 7.0) and centrifuged at $6000 \times g$ for 5 min. The washing step was repeated three times. The pellet (referred to as Si-RP) was redispersed with 250 µL of 1 M PBS and kept at 4 °C until use.

**Immobilization of the capture probe on magnetic beads**. The immobilization of biotinylated CP on magnetic beads (Dynabeads™ MyOne™ Streptavidin T1) was performed according to the manufacturer's instructions. In all, 4 µL of 100 µM CP was mixed with 100 µL of magnetic beads (10 mg/mL) and 12 µL of 100 µM of poly T-BP. The mixture was incubated at room temperature for 30 min. The CP-MNBs were separated from the unbound CP by magnet separation, followed by a washing step of three times with 20 mM PBS (pH 7.0). Finally, the CP-MNB were redispersed with 100 µL of 20 mM PBS and stored at 4 °C until use.

**Multiplex RCA assay**. A step-by-step protocol describing the assay can be found at Protocol Exchange[52]. The RCA assay consists of two steps, which are padlock ligation and RCA amplification[53]. For the ligation reaction, 0.6 µL of 1 µM circular DNA template for each gene and 1.8 µL of target DNA/cDNA/RNA were added to the ligation solution that contained 4 µL of 10× DNA ligase buffer and 1 µL of 5 U/mL T4 DNA ligase. The reaction mixture was incubated at room temperature for 10 min followed by a heat inactivation step at 65 °C for 5 min. The ligation product, called Padlock DNA, served as the template for amplification. For the RCA amplification, the reaction mixture contained 3 µL Padlock DNA, 1 µL dNTPs, 1 µL of 10× phi29 polymerase buffer, 0.1 µL of phi29 DNA polymerase (10 U/mL) in a final volume of 10 µL. The mixture was incubated at 30 °C for 30 min, followed by heat inactivation at 95 °C for 5 min. The mixture containing the RCA amplicons was resolved by agarose gel electrophoresis. In brief, 2 µL of the RCA amplicons were resolved using 0.8% agarose gel in Tris-acetate-EDTA buffer. The resolved gel was visualized under UV light.

**Target hybridization and electrochemical detection**. First, 2 µL of CP-MNB and 18 µL of RCA amplicons were mixed and incubated at 50 °C for 30 min. Then, a magnet was applied to the side of the tube to allow the solid-solution phase separation, followed by washing once with 20 mM PBS-Tween and three times with 20 mM PBS[54]. In all, 20 µL of Si-RP was added to the CP-MNB-Target mixture and incubated at 50 °C for 30 min to allow sandwich hybridization to occur. After that, the pellet was washed three times with 20 mM PBS-Tween and facilitated by magnet separation. Finally, the pellet was resuspended with 0.1 M PB containing potassium chloride (PB/KCl) (pH 7.0). This solution was pipetted onto the SPCE. DPV was performed by scanning from −0.8 to −0.05 V, with a step potential of 0.01 V, modulation amplitude of 0.1 V with an interval time of 0.01 s, and 0.1 V/s scan rate.

Another hybridization strategy, called one-step hybridization, was also tested to shorten the assay time. All components (CP-MNB, Si-RP, and target) were mixed together with the same volume as previously described, and incubated at 50 °C for 30 min. Afterwards, the washing step was performed with 400 µL of 20 mM PBS

(pH 7.0) and the pellet was resuspended in 150 µL of 0.1 M PB/KCl (pH 7.0). This solution was used for DPV measurements.

**Sensitivity and specificity of the assay**. The assay sensitivity was determined using pGEM-T Easy Vector plasmids (Promega, Madison, WI) containing the target sequence of the N (961 bp) and S (1119 bp) genes of SARS-CoV-2. Ten-fold serial dilutions ($1 \times 10^{10}$ to 0.1 copies/µL) of each plasmid and was prepared and 1.8 µL was used as the template in RCA, followed by hybridization and electrochemical detection as described in the previous sections. The assay specificity was evaluated with 1.8 µL of complementary (1 pM) and non-complementary targets (1 pM). Non-complementary targets comprised of short gene fragments of IAV and IVB, and two bases mismatch linear targets.

**Detection of SARS-CoV-2 from clinical samples**. Respiratory samples remaining from diagnostic tests conducted by the Institute for Urban Disease Control and Prevention, Department of Disease Control, Ministry of Public Health Thailand, were used to evaluate the performance of the assay. A total of 106 anonymized respiratory clinical samples were used to evaluate the performance of the assay. The samples include SARS-CoV-2-positive samples and samples that tested positive for other respiratory viruses such as the influenza virus and a respiratory syncytial virus. The SARS-CoV-2-positive samples were obtained in the form of purified RNA from the Institute for Urban Disease Control and Prevention, Department of Disease Control, Ministry of Public Health, Thailand[55]. The RNA was extracted from 200 µL of nasopharyngeal swab sample using a magLEAD Consumable Kit with the magLEAD 12gC instrument (Precision System Science, Chiba, Japan) following the manufacturer's protocol. The SARS-CoV-2-positive samples had been tested and confirmed by the same reference laboratory using the Allplex 2019-nCoV multiplex qRT-PCR assay (Seegene, Seoul, Republic of Korea) with primers and probes specifically targeting the N and S genes.

The RNA specimens were transferred to the Center of Excellence in Clinical Virology at Chulalongkorn University for cDNA synthesis using ImProm-II Reverse Transcription System (Promega, Madison, WI) and random hexamer primers according to the manufacturer's protocol. In all, 51 RNA samples (11 positive and 40 negative for SARS-CoV-2) and 55 cDNA samples (30 positive and 16 negative for SARS-CoV-2, nine negative for SARS-CoV-2 but positive for other respiratory viruses) were used as templates in RCA, followed by electrochemical detection as described in the previous sections.

**Ethics statement**. The study was reviewed and approved by the institutional review board of the Ethics Committee of the Faculty of Medicine, Chulalongkorn University, Thailand (IRB number 301/63). The institutional review board did not require written informed consent because the study samples were anonymous.

**Statistics**. Statistic significances were calculated by Microsoft Excel version 16.30 and all the data were shown as mean ± s.d. The two-tailed Student's *t* test was used to compare differences between two groups with a *P* value < 0.05 as a threshold for significance. Five technical replicates were performed to improve the statistics.

**Reporting summary**. Further information on research design is available in the Nature Research Reporting Summary linked to this article.

## Data availability

The authors declare that the data supporting the findings of this study are available within the paper and its supplementary information files or from the corresponding author upon reasonable request. Sequence information used in this study was from National Center for Biotechnology Information (NCBI) with GenBank accessions of MN908947.3. Source data are provided with this paper.

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

## Acknowledgements

This work was supported by the National Research Council of Thailand, the Research Chair Grant from the National Science and Technology Development Agency (P–15–5004), The authors would like to thank the staffs and nurses of the Institute of Urban Disease Control and Prevention (IUDC), Department of Disease Control, Ministry of Public Health, Thailand for the support and valuable specimens' collection.

## Author contributions

B.L., S.Y.L., N.B., T.N., and T.C. contributed to designing the study, writing, and editing the manuscript. B.L., N.B., and T.C. performed experiments. B.L., T.N., and P.A. analyzed data and prepared figures. A.O.M., S.V., and Y.P. conceived and supervised the study and wrote the manuscript. J.P. obtained ethical approval for this study. All authors participated in manuscript editing and approved the manuscript.

## Competing interests

The authors declare no competing interests.
