## [Peer Review File · Nature Communications]

Reviewers' Comments:

Reviewer #1:

Remarks to the Author:

The manuscript describes the development of electrochemical detection of SARS-CoV-2 viral genes using isothermal rolling circle amplification (RCA). While considered as a standard for COVID-19 diagnosis, the PCR-based detection of SARS-CoV-2 has some major drawbacks such as requirements of PCR instruments and specialized operators. Isothermal amplification-based detection method described in this manuscript utilizes a portable potentiostat device which enables on-site diagnosis of COVID-19 with high sensitivity comparable to the RT-PCR assay.

Although the manuscript has its originality in achieving the diagnosis of COVID-19 targeting multiple genes in point-of-care setting with sufficient diagnostic accuracy, verified with patient samples, the concept of RCA-based viral genosensor, which the author claims to be the first for SARS-CoV-2, has already been published in other journals (Tian et al., *Biosens Bioelectron* 165, 112356 (2020); Huang et al., *bioRxiv* DOI: 10.1101/2020.06.12.128876 (2020)). Furthermore, there are some claims which seems to have insufficient data to support. Therefore, this paper is not appropriate to publish in this journal. The reviewer suggests that the authors should reinforce such claims with additional experiments before the manuscript can be considered for publication.

The major concerns are as follows:

λ In the assessment of assay sensitivity, the authors note that both the N and S genes showed similar current readings although different redox dyes were used for each genes (line 164); however, in-depth discussion for this result is lacked. The reviewer wonders how this result occurred although the gene size and the amplification efficiency are not identical, what this result signifies, and if this result affects the performance of multiplex detection, either positively or negatively.

λ In the sensitivity of assay (line 167), the authors claim that the linear range for detecting both N and S genes are 1 to 1×10^{10} copies/ μ L, with R² value of 0.99. However, judging from Figure 2c or the dataset provided from the authors, the 1×10^{10} copies/ μ L concentration seems to be out of linear range. The author should re-consider the linear range of detection.

λ For quantitative detection of viral genes, the recovery efficiency experiment at various concentration of spiked viral genes should also be conducted for evaluation of the reliability of measurements.

λ In the comparison of RCA-based assay of clinical sample with the result of qRT-PCR (line 210), the authors claim that the sample with lower C_q value (due to higher viral titer) showed higher current signal in RCA-based assay. Although data from Figure 3 generally seems to have this trend, some data points do not follow the same tendency. The reviewer is curious about the reason for these inconsistencies, and if these inconsistencies affect the accuracy of the assay.

λ As mentioned in the discussion section (line 308), and as reported in other literatures, the SARS-CoV-2 viral titer in clinical sample differs with the collecting method of specimen, i.e. nasopharyngeal swabs, throat swabs or sputum. In this manuscript the authors used two types of specimen, nasopharyngeal and throat swabs, to validate the application of their method in clinical settings. However, in Figure 3, the authors merged the two different types of specimen without classification. Although the signal reading seems to appear in "on-and-off" manner, the reviewer does not consider it a good idea to merge the results from two different sample specimen.

The following points should also be amended or supplemented:

λ In Figure 1e, the reviewer consider that the authors should add the electrophoresis result of circular DNA templates in the absence of target gene and without amplification.

λ The unit of target gene concentration is not consistent in the text. For example, Figure 2b describes the N gene concentration as pM, but in Figure 2c the concentration was described in copies/ μ L.

λ In Figure 2d and 2e, it is difficult to identify the concentration range in the plot, since the plot is too cluttered with no legend or description in caption.

λ In the assessment of selectivity (Figure 2g), the author should state the concentration of Influenza A, Influenza B virus genes and two-base mismatch sequence used for experiment.

λ In the validation of performance of RCA-based assay using clinical sample, the authors used both RNA extracts and cDNA samples. For the better understanding of the readers, it is better to note the difference in RNA-based and cDNA-based viral gene detection, and the reason they used both samples for validation.

λ The number of SARS-CoV-2 positive RNA sample in Figure 3a and 3b is eleven, not ten.

λ The authors should check the numbering of the tables in the manuscript: the table regarding the

comparison of RCA-based assay with detection methods in other literatures should be labeled as Table 1.

λ The sample volume used to validate sensitivity and specificity of RCA-based assay is not reported in the method section.

λ For RCA-based assay, all measurement was conducted with five technical replicates. The reviewer wonders if the qRT-PCR assay was also conducted multiple times since there is no error bar present in Figure 3.

λ The information about the sample collection and treatment was not clearly described in the manuscript. The authors should specify where and how the clinical samples were obtained, and whether the study was approved by Ethical Committee.

λ Some of the abbreviations are used without definition, or not defined at the first appearance in the manuscript.

λ Some figures have low resolutions, and the texts in the graphics (such as in Scheme 1, Figure 2a and 2b) are difficult to recognize. The figure in Supplementary Information is also too small and hard to recognize. The authors should consider dividing or rearrange the figures for better visualization.

Reviewer #2:

Remarks to the Author:

Chaibun et al., have developed an electrochemical biosensor that uses rolling circle amplification (RCA) to detect SARS-CoV-2. They report high sensitivity, good selectivity, and a broad linear range. Overall this approach seems interesting and potentially useful even if it isn't practical in its current form due to constraints with opening tubes post amplification and washing. I believe the results from this can built on by others to potentially reach a truly fieldable solution.

Comments:

Intro – I wouldn't be quite so critical of serological assays. They have a role and the good ones can be pretty accurate. Also, antigen assays should be mentioned.

The methods need to be clearer when it comes to the assay and the samples. It was quite hard for me to figure out exactly what the true sensitivity of the assay was. 0.1/uL is impressive or not depending on how many uLs are used. I would ideal want to see how many molecules are actually in each assay. Given the error bars at 0.1/uL there must be many uL per sample. It would also be important to show some calibration that shows that the x-axis is accurate (e.g. qPCR).

To really assess how well this assay works, there should be side-by-side data with RT-qPCR so we can better understand 1) the viral titer of the samples chosen and 2) assess sensitivity (i.e. we need an LoD curve of this assay side-by-side with RT-qPCR).

It would nice to see the assay tried on a crude lysate as that would greatly simplify the work flow. I don't consider this essential nor would it not working on crude lysate effect whether I thought this was suitable for publication, but it would be nice to know.

Minor:

Fig 1a. scale bar should be "µm" instead of "µM"

Fig 1e. Should include a negative control (e.g. conditions with RCA on non-target genes)?

Fig. 2c. Where is the blank (control background) signal threshold? Please mark it.

Fig. 2c. Is this RNA or DNA input? Please specify.

Fig. 2d and e. Where is the no template control line? Please mark it.

Fig. 2g. How much IAV and IBV input are you testing? Please show a titration curve with IAV and IBV as input and use N and S gene probes.

Dear Editor,

We thank the reviewers for their critical assessment of our work. We have addressed each of the comments raised by the reviewers below and incorporated their feedback into the revised manuscript which is highlighted in yellow. We believe that this has improved the quality of the manuscript which we now hope is suitable for publication in Nature Communication.

Response to reviewers

Reviewer 1:

The manuscript describes the development of electrochemical detection of SARS-CoV-2 viral genes using isothermal rolling circle amplification (RCA). While considered as a standard for COVID-19 diagnosis, the PCR-based detection of SARS-CoV-2 has some major drawbacks such as requirements of PCR instruments and specialized operators. Isothermal amplification-based detection method described in this manuscript utilizes a portable potentiostat device which enables on-site diagnosis of COVID-19 with high sensitivity comparable to the RT-PCR assay.

Although the manuscript has its originality in achieving the diagnosis of COVID-19 targeting multiple genes in point-of-care setting with sufficient diagnostic accuracy, verified with patient samples, the concept of RCA-based viral genosensor, which the author claims to be the first for SARS-CoV-2, has already been published in other journals (Tian et al., Biosens Bioelectron 165, 112356 (2020); Huang et al., bioRxiv DOI: 10.1101/2020.06.12.128876 (2020)). Furthermore, there are some claims which seems to have insufficient data to support. Therefore, this paper is not appropriate to publish in this journal. The reviewer suggests that the authors should reinforce such claims with additional experiments before the manuscript can be considered for publication.

We thank the reviewer for raising this point. Although the studies by Tian et al. and Huang et al. used RCA for SARS-CoV-2 detection, they have performed the testing with synthetic DNA targets. In our paper, we have tested the RCA with RNA and cDNA from clinical samples. Thus, we have revised our earlier statement as “To the best of our knowledge, our study is the first report of an RCA-based electrochemical biosensor assay for the detection of SARS-CoV-2 from clinical samples (Line 90). We have also added the study by Tian et al. in the introduction (Line 89) as follows: “Recently, an RCA-based real-time optomagnetic detection of synthetic

complementary DNA of SARS-CoV-2 was reported.” We did not include the paper by Huang et al. as it is only available as a pre-print online.

1) *In the assessment of assay sensitivity, the authors note that both the N and S genes showed similar current readings although different redox dyes were used for each genes (line 164); however, in-depth discussion for this result is lacked. The reviewer wonders how this result occurred although the gene size and the amplification efficiency are not identical, what this result signifies, and if this result affects the performance of multiplex detection, either positively or negatively.*

We thank the reviewer for the suggestion and for highlighting this point. In this work, we measured the current signal from the oxidation of redox dyes labelled on DNA reporter probes. The silica (Si)-redox dye had been prepared separately for each dye (methylene blue (MB) and acridine orange (AO) using the same concentration of redox dye, which is 15 μM on 0.3 g of silica. The thermodynamic and chemical interactions of these 2 silica dyes should be the same and therefore the surface coverage of each dye should be very similar. Both dyes involve a 2 electron oxidation process¹ and given that the electrochemical response is a surface confined process the loading of the dye will be the most critical factor, which being the same will result in similar responses. To demonstrate this further we constructed the experiment to measure the current from Si-MB and Si-AO at the same dilution and the signal with 3 replicates were plotted and shown in the figures below.

Differential pulse voltammograms and bar graphs showing the current signal of SiMB (purple line) and SiAO (orange line)

For the assessment of assay sensitivity, the similarity in the current reading for the same concentration of plasmid DNA target could be due to both genes having equal amplification and detection efficiency. We designed the circular DNA for both genes with the same length (66 bases) and same sequence, except for the reporter probe and target gene binding sequences

(see Fig 1b). Although the reporter probe and target gene binding sequences were different for both genes, we ensured that the sequences have almost the same GC content (47-50%). The same universal capture probe was used for both genes. Since the assay is a multiplex detection, an even amplification and detection efficiency of both target genes are desired.

1. Kinoshita T, Hatsuoka Y, Nguyen DQ, Iwata R, Shiigi H, Nagaoka T. Electrochemical Response of Acridine Orange in Bacterial Cell. *Electrochemistry* **84**, 334-337 (2016).

2) In the sensitivity of assay (line 167), the authors claim that the linear range for detecting both N and S genes are 1 to 1×10^{10} copies/ μ L, with R^2 value of 0.99. However, judging from Figure 2c or the dataset provided from the authors, the 1×10^{10} copies/ μ L concentration seems to be out of linear range. The author should re-consider the linear range of detection.

We thank the reviewer for pointing this out. We have amended the sentence in line 178 as follows "The detection limit of both the N and S genes was 1 copy/ μ L, with a linear range of 1 to 1×10^9 copies/ μ L".

3) For quantitative detection of viral genes, the recovery efficiency experiment at various concentration of spiked viral genes should also be conducted for evaluation of the reliability of measurements.

We thank the reviewer for this suggestion. In Thailand, it is protocol that all SARS-CoV-2 sample handling and RNA extraction are performed by the Institute for Urban Disease Control and Prevention, Department of Disease Control, Ministry of Public Health, Thailand. The RNA extraction was performed using an automated system as described in our Methods section. We obtained the RNA extracts from the Department for our study, as our labs are not allowed to handle the clinical samples directly. Therefore, we are unfortunately unable to perform the spiked viral genes for recovery efficiency experiment due to these strict restrictions.

4) In the comparison of RCA-based assay of clinical sample with the result of qRT-PCR (line 210), the authors claim that the sample with lower C_q value (due to higher viral titer) showed higher current signal in RCA-based assay. Although data from Figure 3 generally seems to have this trend, some data points do not follow the same tendency. The reviewer is curious about the reason for these inconsistencies, and if these inconsistencies affect the accuracy of the assay.

We agree with the reviewer and have revised the sentence (line 222) as follows “This result was concordant with the qRT-PCR results.”

The small discrepancies of some results from the qRT-PCR tests and our assay are likely since the clinical samples collection and RNA extraction were done between February – April 2020 (Peunpa et.al., 2020, reference 57) and the qRT-PCR were done without replication. It is possible that some of the RNA extracts could have been slightly degraded when they were provided to us for this study. However, due to the high sensitivity of our assay, the presence of low copies of the target genes could still be detected.

5) As mentioned in the discussion section (line 308), and as reported in other literatures, the SARS-CoV-2 viral titer in clinical sample differs with the collecting method of specimen, i.e. nasopharyngeal swabs, throat swabs or sputum. In this manuscript the authors used two types of specimen, nasopharyngeal and throat swabs, to validate the application of their method in clinical settings. However, in Figure 3, the authors merged the two different types of specimen without classification. Although the signal reading seems to appear in “on-and-off” manner, the reviewer does not consider it a good idea to merge the results from two different sample specimen.

We apologize for the confusion. The RNA samples provided to us were extracted from nasopharyngeal swabs, not from both nasopharyngeal and throat swabs. We have amended the type of clinical sample used for evaluation as nasopharyngeal swab in line 499, Figure 3 caption and Table 1.

The following points should also be amended or supplemented:

6) In Figure 1e, the reviewer consider that the authors should add the electrophoresis result of circular DNA templates in the absence of target gene and without amplification.

We thank the reviewer for this suggestion. We have added “The electrophoresis result of circular DNA templates in the absence of target gene and without amplification is shown in Supplementary Fig. 1.” (line 136) as seen below. In Lane 2, a band of the circular DNA template

(66 bases) can be seen below the 100 bp band. Lanes 3 and 4 show the bands of Sg and Ng DNA linear targets. Lane 5 shows the Padlock DNA band around 100 bp. Although Padlock DNA and circular DNA template are both 66 bases, the circular form of the Padlock DNA causes it to migrate slower than the “linear” circular DNA (Lane 2). Lane 6 and 7 show bands with the same size as circular DNA template (Lane 2) because in the absence of Sg-LT, the Padlock DNA does not form. Lane 8 shows a band at the well corresponding to the high molecular weight RCA amplicons. Lanes 9 and 10 do not show any band as no RCA reaction / amplicons were formed.

Lane 1 – Ladder 100 bp
 Lane 2 - Circular DNA template for S gene
 Lane 3 - DNA linear target for S genes
 Lane 4 - DNA linear target for N genes
 Lane 5 – Sg Circular DNA template with Sg-LT
 Lane 6 – Sg Circular DNA template with non-complementary (NC)
 Lane 7 – Sg Circular DNA template with phosphate buffer (PB)
 Lane 8 – RCA with Padlock DNA reaction from Lane 5
 Lane 9 – RCA with reaction from Lane 6
 Lane 10 – RCA with reaction from Lane 7

7) *The unit of target gene concentration is not consistent in the text. For example, Figure 2b describes the N gene concentration as pM, but in Figure 2c the concentration was described in copies/ μ L.*

We would like to clarify the differences in the unit of target gene concentration used in Fig. 2b and Fig. 2c. In Fig. 2b, the N gene used for optimization of hybridization was a short synthetic linear DNA, which we expressed in ‘pM’, whereas the N gene used for the assay sensitivity was a plasmid cloned with the N gene, which we expressed in ‘copies/ μ L’. For better comparison of

the assay sensitivity, we have also shown the plasmid concentrations in molarity in Supplementary Table 1.

8) In Figure 2d and 2e, it is difficult to identify the concentration range in the plot, since the plot is too cluttered with no legend or description in caption.

We thank the reviewer for pointing that out. We have included the concentration range beside the arrow in the plots, as shown below:

9) In the assessment of selectivity (Figure 2g), the author should state the concentration of Influenza A, Influenza B virus genes and two-base mismatch sequence used for experiment.

We have stated the concentration of the non-complementary targets as 1 pM in the revised manuscript as follows:

In line 186: "In the specificity testing, 1 pM of target sequences of Influenza A (IAV) and Influenza B (IVB) viruses were included as non-complementary targets, where the sequence alignment is shown in Fig. 2f. In addition, 1 pM of linear targets with two bases mismatch were also included."

In line 488: "The assay specificity was evaluated with 1.8 μL of complementary (1 pM) and non-complementary targets (1 pM). Non-complementary targets comprised of short gene fragments of Influenza A (IAV) (and Influenza B (IVB) viruses, and 2-bases mismatch linear targets".

10) In the validation of performance of RCA-based assay using clinical sample, the authors used both RNA extracts and cDNA samples. For the better understanding of the readers, it is better to note the difference in RNA-based and cDNA-based viral gene detection, and the reason they used both samples for validation.

Our aim is to perform direct amplification and detection from RNA samples, which we managed to achieve in this study. We included cDNA samples as cDNA is more stable and comparatively easier to handle compared to the RNA template.

11) The number of SARS-CoV-2 positive RNA sample in Figure 3a and 3b is eleven, not ten.

We apologize for the mistake. The number of SARS-CoV-2 positive RNA sample is supposed to be 11, not 10, as stated in line 219 and 508. The total number of SARS-CoV-2 positive samples (RNA + cDNA) is 41 and the total number of clinical samples used is 106. We have amended the mistakes in the following sentences:

Abstract, line 29: Sensor evaluation with 106 clinical samples, including 41 SARS-CoV-2 positive and 9 samples positive for other respiratory viruses, gave a 100% concordance result with qRT-PCR, with complete correlation between the biosensor current signals and quantitation cycle (C_q) values.

Results, line 219: All 41 samples (11 RNA and 30 cDNA) prepared from SARS-CoV-2 positive clinical samples yielded positive results, while the 65 samples (40 RNA, 25 cDNA) prepared from SARS-CoV-2 negative clinical samples recorded negative results (Fig. 3).

Discussion, line 307: The application of the optimized assay for the diagnosis of COVID-19 was evaluated with 106 clinical samples.

Methods, line 508: Fifty one RNA samples (11 positive and 40 negative for SARS-CoV-2) and 55 cDNA samples (30 positive and 16 negative for SARS-CoV-2, 9 negative for SARS-CoV-2 but positive for other respiratory viruses) were used as templates in RCA, followed by electrochemical detection.

12) The authors should check the numbering of the tables in the manuscript: the table regarding the comparison of RCA-based assay with detection methods in other literatures should be labeled as Table 1.

We apologize for the mistake. This has been amended as Table 1 in the revised manuscript.

13) The sample volume used to validate sensitivity and specificity of RCA-based assay is not reported in the method section.

We have included the sample volume used to validate sensitivity and specificity (line 486) as follows: "Ten-fold serial dilutions (1×10^{10} to 0.1 copies/ μL) of each plasmid and was prepared and 1.8 μL was used as the template in RCA, followed by hybridization and electrochemical detection as described in the previous sections. The assay specificity was evaluated with 1.8 μL of complementary (1 pM) and non-complementary targets (1 pM)."

14) For RCA-based assay, all measurement was conducted with five technical replicates. The reviewer wonders if the qRT-PCR assay was also conducted multiple times since there is no error bar present in Figure 3.

According to the Institute for Urban Disease Control and Prevention, Department of Disease Control, Ministry of Public Health, Thailand who provided us the RNA extracts and Cq values of the qRT-PCR, the qRT-PCR assay was conducted once for each sample, using two separate multiplex RT-PCR assays (Puenpa et al., 2020, reference 57). In the first multiplex assay, the Allplex 2019-nCoV (Seegene, Seoul, Republic of Korea) incorporating primers and probes specifically targeting RdRp, N and E genes was used. The second assay used the LightMix Modular SARS and Wuhan CoV (TIB-Molbiol, Berlin, Germany) incorporating primers and probes corresponding to the RdRp and E genes. We were provided the Cq values for the first multiplex assay using Allplex 2019-nCoV.

15) The information about the sample collection and treatment was not clearly described in the manuscript. The authors should specify where and how the clinical samples were obtained, and whether the study was approved by Ethical Committee.

We have added the reference Puenpa et al., 2020 (reference 57) that describes the information about the sample collection, treatment and ethical approval in the revised manuscript (line 493-518). The SARS-CoV-2 strains were collected from the outbreak in Thailand during the first wave from February to April 2020. The specimens were collected from various locations in Thailand such as Bangkok, Nonthaburi, Samut Prakan, Songkla, Ubon Ratchathani and Yala. Nasopharyngeal samples were obtained in viral transport medium and RNA extraction was performed using using a magLEAD 12gC instrument (Precision System Science, Chiba, Japan) with a magLEAD Consumable Kit (Precision System Science, Chiba, Japan) in accordance with the manufacturer's instructions.

Patient identifiers including personal information and hospitalization number were removed from the samples to ensure patient confidentiality. The research proposal was approved by the institutional review board of the Ethics Committee of the Faculty of Medicine, Chulalongkorn University, Thailand (IRB number 301/63). The institutional review board of the Ethics Committee for human research waived the need for consent because all samples were anonymous. All methods were performed in accordance with the relevant guidelines and regulations.

16) Some of the abbreviations are used without definition, or not defined at the first appearance in the manuscript.

We have checked the manuscript and included the definition at the first appearance in the revised manuscript.

17) Some figures have low resolutions, and the texts in the graphics (such as in Scheme 1, Figure 2a and 2b) are difficult to recognize. The figure in Supplementary Information is also too small and hard to recognize. The authors should consider dividing or rearrange the figures for better visualization.

We thank the reviewer for the suggestion. We have revised the figures by improving its

resolution.

Reviewer #2 (Remarks to the Author):

Chaibun et al., have developed an electrochemical biosensor that uses rolling circle amplification (RCA) to detect SARS-CoV-2. They report high sensitivity, good selectivity, and a broad linear range. Overall this approach seems interesting and potentially useful even if it isn't practical in its current form due to constraints with opening tubes post amplification and washing. I believe the results from this can built on by others to potentially reach a truly fieldable solution.

Comments:

1) Intro – I wouldn't be quite so critical of serological assays. They have a role and the good ones can be pretty accurate. Also, antigen assays should be mentioned.

We thank the reviewer for the suggestion. We have included a paragraph about serological tests in line 53 as follows:

“There are two types of diagnostic tests for COVID-19, serological and viral nucleic acid tests. Serological testing detects the presence of antibodies produced by an individual due to exposure to the virus or detection of antigenic viral proteins in the infected individuals. An antibody test should not be used to diagnose someone with an active infection and may give false-negative results. Rapid antigen tests for SARS-CoV-2 are relatively inexpensive and give immediate results. However, these tests are generally less sensitive than nucleic acid-based tests. At the early stage of infection, the human immune system may not be active and thus cause false-negative diagnosis². “

2) The methods need to be clearer when it comes to the assay and the samples. It was quite hard for me to figure out exactly what the true sensitivity of the assay was. 0.1/uL is impressive or not depending on how many uLs are used. I would ideal want to see how many molecules are actually in each assay. Given the error bars at 0.1/uL there must be many uL per sample.

The volume used for testing is mentioned in the Methods section. For evaluation of assay sensitivity, we used 1.8 μL of plasmid DNA with concentrations ranging from 0.1 – 10^{10} copies/ μL (see line 486). As we mentioned in line 176 of Results “The detection limit of both the N and S genes was 1 copy/ μL , with a linear range of 1 to 1×10^9 copies/ μL .” Therefore, the assay could detect 1.8 copy of the N or S gene per 1.8 μL of plasmid DNA sample used.

3) It would also be important to show some calibration that shows that the x-axis is accurate (e.g. qPCR). To really assess how well this assay works, there should be side-by-side data with RT-qPCR so we can better understand 1) the viral titer of the samples chosen and 2) assess sensitivity (i.e. we need an LoD curve of this assay side-by-side with RT-qPCR).

We thank the reviewer for the suggestion. For the validation with clinical samples, the RNA extraction and RT-qPCR were performed by the Institute for Urban Disease Control and Prevention, Department of Disease Control, Ministry of Public Health, Thailand. We were only provided the Cq data as the RT-qPCR test results were qualitative. Therefore, unfortunately we do not have the information of viral titer in the clinical samples to do a side-by-side comparison of sensitivity.

4) It would nice to see the assay tried on a crude lysate as that would greatly simplify the work flow. I don't consider this essential nor would it not working on crude lysate effect whether I thought this was suitable for publication, but it would be nice to know.

We agree with the reviewer on this point. However, due to strict regulations on handling SARS-CoV-2 samples in Thailand, our lab is not allowed to handle crude lysate.

Minor:

5) Fig 1a. scale bar should be “ μm ” instead of “ μM ”

Fig 1a scale bar has been corrected to “ μm ”.

6) Fig 1e. Should include a negative control (e.g. conditions with RCA on non-target genes)?

We have added “The electrophoresis result of circular DNA templates in the absence of target gene and without amplification is shown in Supplementary Fig. 1.” (line 136) as seen below. In Lane 2, a band of the circular DNA template (66 bases) can be seen below the 100 bp band. Lanes 3 and 4 show the bands of Sg and Ng DNA linear targets. Lane 5 shows the Padlock DNA band around 100 bp. Although Padlock DNA and circular DNA template are both 66 bases, the circular form of the Padlock DNA causes it to migrate slower than the “linear” circular DNA (Lane 2). Lane 6 and 7 show bands with the same size as circular DNA template (Lane 2) because in the absence of Sg-LT, the Padlock DNA does not form. Lane 8 shows a band at the well corresponding to the high molecular weight RCA amplicons. Lanes 9 and 10 do not show any band as no RCA reaction/ amplicons were formed.

Lane 1 – Ladder 100 bp
Lane 2 - Circular DNA template for S gene
Lane 3 - DNA linear target for S genes
Lane 4 - DNA linear target for N genes
Lane 5 – Sg Circular DNA template with Sg-LT
Lane 6 – Sg Circular DNA template with non-complementary (NC)
Lane 7 – Sg Circular DNA template with phosphate buffer (PB)
Lane 8 – RCA with Padlock DNA reaction from Lane 5
Lane 9 – RCA with reaction from Lane 6
Lane 10 – RCA with reaction from Lane 7

7) *Fig. 2c. Where is the blank (control background) signal threshold? Please mark it.*

Fig. 2c has been revised to include the blank (control background) signal.

8) *Fig. 2c. Is this RNA or DNA input? Please specify.*

Fig. 2c shows the sensitivity assay for N and S genes performed with plasmid DNA, as described in Methods section (line 483).

9) *Fig. 2d and e. Where is the no template control line? Please mark it.*

The no template control lines in Fig 2d and e are indicated as dashed lines.

10) *Fig. 2g. How much IAV and IBV input are you testing? Please show a titration curve with IAV and IBV as input and use N and S gene probes.*

1 pM of IAV and IBV were used for testing, as described in Methods section (line 483). As shown in Fig 2g, when IAV and IBV were used as input with N and S gene probes in the multiplex RCA and detection, the current signals for IAV, IBV and IAV + IBV were similar to the blank.

Reviewers' Comments:

Reviewer #1:

Remarks to the Author:

The revised manuscript supplemented many points that had previously mentioned in the review. Although the quality of the manuscript had been improved from the revision, some points need further explanations in the revised manuscript.

1) For the question about the similar signal intensity from N and S genes (question 1 from Reviewer 1), the authors responded that the similarity of the signals was attributed to the same mechanism of the redox reaction (two electron oxidation) for both dye used in this study, and the equal amplification efficiency of both genes due to the same gene length and GC contents. The authors supplemented the measurement of differential pulse voltammograms for both probes without DNA reporter label. Also, the authors noted that the similar signal reading is desired for target genes, since the system aims for multiplex detection. However, these explanation was not included in the revised manuscript. The reviewer suggests to include the explanation in the main text or in supporting information.

2) Also for the answer about the first question, the authors premised that the concentration of both methylene blue and acridine orange are same on the Si probe. However, the reviewer wonders that the number of attached redox dyes on Si probe differ due to the different chemical properties of two dyes. The reviewer suggests to provide a reference that indicates the same incorporation efficiency for both dyes.

3) In the new Supplementary Figure 1, what is the sequence of non-complementary strand?

Reviewer #2:

Remarks to the Author:

The authors have satisfied my concerns.

Point-by-point response to the reviewers' comments

Reviewer #1 (Remarks to the Author):

The revised manuscript supplemented many points that had previously mentioned in the review. Although the quality of the manuscript had been improved from the revision, some points need further explanations in the revised manuscript.

1) For the question about the similar signal intensity from N and S genes (question 1 from Reviewer 1), the authors responded that the similarity of the signals was attributed to the same mechanism of the redox reaction (two electron oxidation) for both dye used in this study, and the equal amplification efficiency of both genes due to the same gene length and GC contents. The authors supplemented the measurement of differential pulse voltammograms for both probes without DNA reporter label. Also, the authors noted that the similar signal reading is desired for target genes, since the system aims for multiplex detection. However, these explanation was not included in the revised manuscript. The reviewer suggests to include the explanation in the main text or in supporting information.

Authors' response:

We thank the reviewer for the suggestion and highlighting the point. We have included the explanation as mentioned in the revised manuscript (See line 177) and Supplementary Fig.2.

2) Also for the answer about the first question, the authors premised that the concentration of both methylene blue and acridine orange are same on the Si probe. However, the reviewer wonders that the number of attached redox dyes on Si probe differ due to the different chemical properties of two dyes. The reviewer suggests to provide a reference that indicates the same incorporation efficiency for both dyes.

Authors' response:

We thank the reviewer for the suggestion. We have included the related references in the revised manuscript (See line 178) and Supplementary Fig.2.

3) In the new Supplementary Figure 1, what is the sequence of non-complementary strand?

Authors' response:

Supplementary Figure 1 is the gel image of the RCA reaction for *S* gene. We have used the *N* gene DNA linear target (Ng-LT) as the non-complementary strand. The sequence of Ng-LT is provided in Table 2. We have revised the description for Supplementary Figure 1 as follows:

Supplementary Figure 1 shows the gel image that visualized the RCA reaction for the *S* gene. In Lane 2, a band of the circular DNA template (66 bases) can be seen below the 100 bp band. Lanes 3 and 4 show the bands of Sg and Ng DNA linear targets. Lane 5 shows the Padlock DNA band around 100 bp, which confirmed the hybridization of circular DNA template and its target gene (*S* gene). Although Padlock DNA and circular DNA template are both 66 bases, the circular form of the Padlock DNA causes it to migrate slower than the “linear” circular DNA (Lane 2). Lane 6 and 7 show bands with the same size as circular DNA template (Lane 2) because in the absence of Sg-LT, the Padlock DNA does not form. Lane 8 shows a band at the well corresponding to the high molecular weight RCA amplicons, which confirmed the specific RCA reaction. Lanes 9 and 10 do not show any band as no RCA reaction / amplicons were formed.